# Transplanted photoreceptor precursors transfer proteins to host photoreceptors by a mechanism of cytoplasmic fusion

Mandeep S. Singh[1],*, Jasmin Balmer[1],*, Alun R. Barnard[1], Sher A. Aslam[1,2], Daniela Moralli[3], Catherine M. Green[3], Alona Barnea-Cramer[1], Isabel Duncan[1] & Robert E. MacLaren[1,2,4]

Photoreceptor transplantation is a potential future treatment for blindness caused by retinal degeneration. Photoreceptor transplantation restores visual responses in end-stage retinal degeneration, but has also been assessed in non-degenerate retinas. In the latter scenario, subretinal transplantation places donor cells beneath an intact host outer nuclear layer (ONL) containing host photoreceptors. Here we show that host cells are labelled with the donor marker through cytoplasmic transfer—94 ± 4.1% of apparently well-integrated donor cells containing both donor and host markers. We detect the occurrence of Cre-Lox recombination between donor and host photoreceptors, and we confirm the findings through FISH analysis of X and Y chromosomes in sex-discordant transplants. We do not find evidence of nuclear fusion of donor and host cells. The artefactual appearance of integrated donor cells in host retinas following transplantation is most commonly due to material transfer from donor cells. Understanding this novel mechanism may provide alternate therapeutic strategies at earlier stages of retinal degeneration.

[1] Nuffield Laboratory of Ophthalmology, Department of Clinical Neurosciences, Levels 5-6 West Wing, University of Oxford, John Radcliffe Hospital, Headley Way, Oxford OX3 9DU, UK. [2] UK Ministry of Defence Army Medical Services, London SW1A 2HB, UK. [3] Wellcome Trust Centre for Human Genetics, Roosevelt Drive, Oxford OX3 7BN, UK. [4] Moorfields Eye Hospital NHS Foundation Trust, 162 City Road, London EC1V 2PD, UK. * These authors contributed equally to this work. Correspondence and requests for materials should be addressed to R.E.M. (email: enquiries@eye.ox.ac.uk).

Studies using stem cell-derived or primary donor cells have shown that transplanting photoreceptors into the degenerate retina improves vision in a variety of animal models of human inherited retinal degenerations (IRD)[1–6]. When the host outer nuclear layer (ONL) has totally degenerated, exogenous photoreceptor replacement results in the *de novo* regeneration of this layer and the restoration of visual function[6]. The results of studies in which primary donor cells were transplanted into hosts that had limited degeneration[5,7], however, made us question whether the appearance of apparently perfectly integrated fluorescently labelled photoreceptors could actually be the host cells, labelled by an artefact of cell fusion. Since there is no known mechanism for photoreceptor cells to 'migrate' into the outer retina, we hypothesized that stable contact might allow fluorescent subretinal cells to label host photoreceptors retrogradely—similar to the known fusion that occurs between photoreceptor outer segment discs and the retinal pigment epithelium. We also considered the evidence that after transplantation of primary or stem cell-derived photoreceptor precursors, fluorescence-tagged cells detected in the host outer nuclear layer (ONL) were reported to 'adopt the morphology' of the host cells[3,5,8].

Here we show that following photoreceptor transplantation, host photoreceptor cells take up cytoplasmic material from donor photoreceptor precursors, and vice-versa. We found cytoplasmic fusion to be a common event, whereas there was no evidence of fusion between donor and host photoreceptor nuclei. The presence of donor-derived fluorescence in the host ONL could be interpreted as evidence of donor cell migration into the host retina, however our data indicate that in the majority of cases, donor cell nuclei remain in the subretinal space and donor-derived cytoplasm is transferred into host photoreceptor cells. These data call for a re-evaluation of previous data on photoreceptor transplantation and highlight a novel regenerative mechanism which could be used as therapy for blindness due to retinal degenerative diseases.

## Results

### Co-localization of donor and host cytoplasmic markers.
To study the occurrence of intercellular transfer of components between donor and host photoreceptor cells, we transplanted FACS-sorted *Nrl-GFP* donor photoreceptor precursors into the subretinal space of adult host mice in which DsRed was expressed ubiquitously (CAG-DsRed)[9]. In the donor, green fluorescent protein (GFP) expression is restricted to post-mitotic rod precursor cells[10], so any GFP detected in the host ONL following transplantation would have originated from donor rods. Two weeks after transplantation, we detected extensive cytoplasmic co-localization of DsRed and GFP in cells located in the host ONL (Fig. 1a–j). We found that $93.8 \pm 4.1\%$ (mean $\pm$ s.e.m., $n = 371$ cells from three eyes) of morphologically normal GFP + photoreceptor cells in the host retina also co-localized DsRed. To confirm this observation through quantitative analysis, we computed the Mander's overlap coefficient (MOC)[11] to measure the co-distribution of GFP and DsRed in perinuclear photoreceptor cytoplasm in the host ONL. We found a median MOC of 0.9 (range, 0.5–1, $n = 230$ cells in three transplanted eyes; Fig. 1k–m). As further confirmation, we performed gender-mismatched transplants of P3 female *Nrl-GFP* donor photoreceptors into the subretinal space of adult wild-type male hosts. Using X and Y chromosome-specific fluorescent *in situ* hybridization (FISH) probes, we found cytoplasmic GFP, and Y-positive nuclei, in the same cells located in the host outer nuclear layer (Fig. 2a,b). Thus photoreceptor cells in the ONL that are indisputably those of the male host (Y-positive) also contain a cytoplasmic marker from the donor (GFP). In this

experiment, we did not detect polyploidy. Hence we concluded that the fusion event was cytoplasmic rather than nuclear, because the donor cell nucleus remained in the subretinal space.

### Fusion with cells other than rod photoreceptors.
We next explored the question of whether donor cells other than developing rods could undergo fusion with mature host photoreceptor cells. Donor photoreceptor precursors were obtained by dissociating retinas from CAG-DsRed mice that were also homozygous for the *rd1* mutation that were aged one to three postnatal days (P1–3). These were transplanted into the subretinal space of adult *Nrl-GFP* mice. The *Rd1* mice have a rapid degeneration due to deficiency of beta-6 phosphodiesterase (PDE) in which rods begin to degenerate at P8 and are completely lost by P20 (refs 12,13). We hypothesized that donor rods would not be detectable in the host ONL if transplanted when aged P3 and assayed histologically three weeks later, that is, at P24 donor cell age. Surprisingly, we detected DsRed + photoreceptor cells with normal morphology, co-localizing GFP, in the host ONL (Fig. 3a–d) 3 weeks following transplantation. As DsRed was expressed in all other donor cell types (cone photoreceptors, Muller glia or other retinal cells), these could have been the source of DsRed protein that was seen in the host ONL. These findings could be explained by mature host rod photoreceptors that had fused with donor cells other than rod precursors, resulting in cytoplasmic transfer of proteins expressed exclusively by the donor cells. Alternatively, it is also possible that donor rod precursors could have been rescued from degeneration by the cytoplasmic transfer of beta-6 PDE in the reverse direction of the DsRed—from the host rod outer segments into the maturing *rd1* mouse donor cells.

### Assessment of bi-directional cytoplasmic transfer.
To understand better the directional nature of cell fusion we explored the possibility of a Cre-lox recombination event between donor and host photoreceptors that was dependent on the transfer of protein in the opposite direction, from host to donor. We therefore generated donor mice by crossing the Ai9 Cre reporter strain (*CAG-LSL-tdTomato*) with the Nrl-GFP line. The progeny from this cross have *GFP*-positive donor rod photoreceptors containing a conditional tdTomato Cre reporter allele. Hence, the exposure of these cells to Cre recombinase would lead to tdTomato reporter expression in addition to GFP. We transplanted postnatal day 7 *Ai9, Nrl-GFP* donor cells into *Crx-Cre* host mice in which Cre expression was controlled by the cone–rod homeobox gene promoter and hence was restricted to host photoreceptor cells. We detected the occurrence of dual labelling with GFP and tdTomato (1.3% of GFP + cells counted) in the host ONL as soon as 3 days post transplantation (Fig. 3e–h). This outcome relies on two steps; first the transfer of Cre recombinase from host to donor cells to activate tdTomato expression and second, the transfer back of both tdTomato and GFP from donor into host cell (Fig. 3e–h). However the number of the GFP + host cells that also contained tdTomato was small, suggesting that the fusion event may not be stable. This is because an undetermined time would be required for the Cre-lox recombination event, whereas the GFP transfer could occur almost spontaneously after cytoplasmic fusion. It cannot, however, be excluded that this occurred by two successive fusion events, with Cre recombinase being taken up from one photoreceptor and subsequent fusion and retrograde labelling of another.

The current data indicate that when photoreceptors are transplanted into the subretinal space and come into contact with the host ONL, cell fusion between donor and host facilitates intercellular exchange of cytoplasmic components. In this context, host photoreceptors may become labelled with the donor fluorescent marker. This can give rise to an artefactual

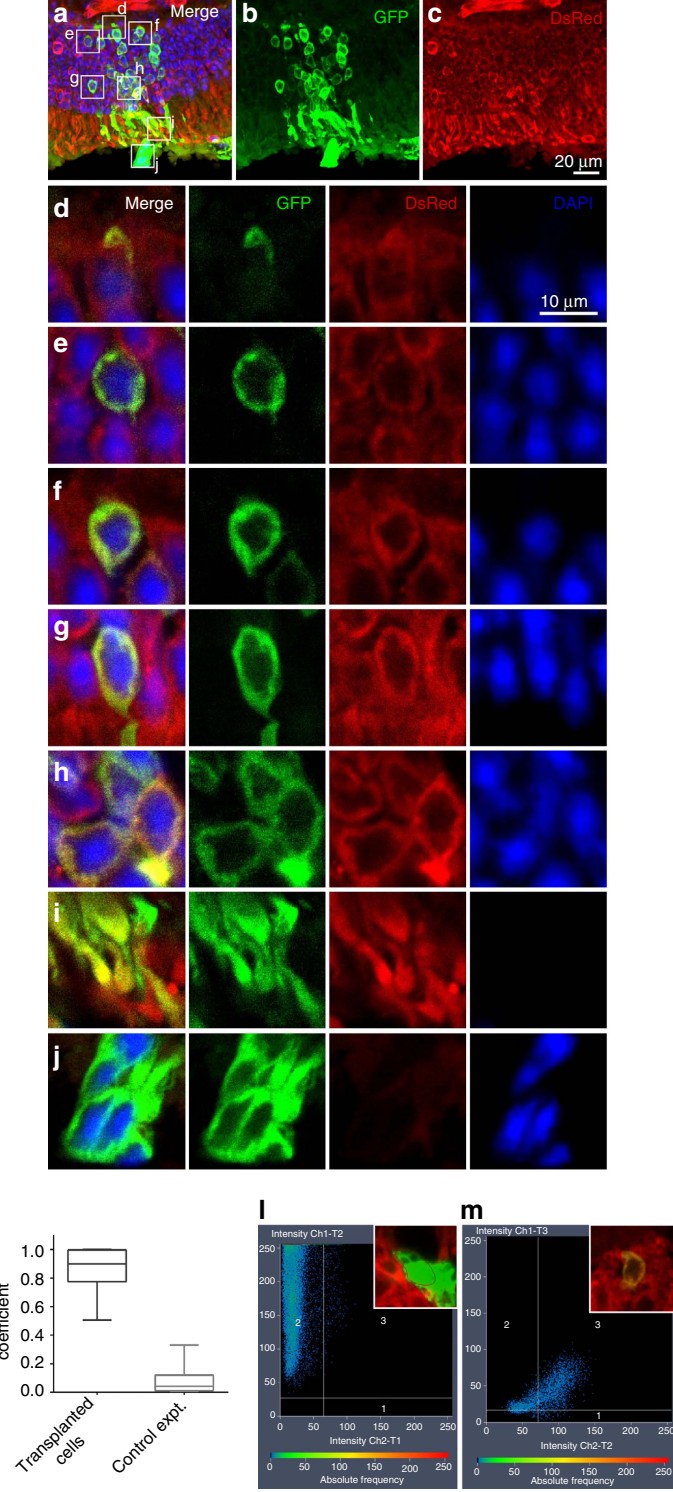

**Figure 1 | The majority of putative donor cells (GFP positive) in the host outer nuclear layer also contain a fluorescent marker that is present in the host (DsRed).** Postnatal day 1–3 (P1–3) *Nrl-GFP* donor photoreceptor precursors were transplanted into the subretinal space of adult *CAG-DsRed* mice without retinal degeneration and after 3 weeks, numerous GFP-positive photoreceptors were found in the host outer nuclear layer. (**a**) Shows merged GFP, DsRed and DAPI channels, (**b**) GFP only and (**c**) DsRed only. GFP and DsRed were co-distributed in cytoplasm in the (**d**) rod spherule synapse, (**e**–**h**) perinuclear cytoplasm and (**i**) inner segments. (**j**) By comparison, the majority of donor photoreceptors cells in the subretinal space contained GFP but were negative for DsRed. (**k**) The mean Mander's overlap coefficient (MOC) of DsRed and GFP in cells located in the ONL was 0.9. The horizontal lines indicate the medians, the boxes extend from the 25th to 75th percentiles and the whiskers indicate the minimum and maximum values. (**l**) A GFP+ cell located outside the host outer nuclear layer (ONL) with low DsRed and GFP co-localization (MOC = 0.059). (**m**) A cell located in the host ONL with a high degree of co-localization (MOC = 0.96).

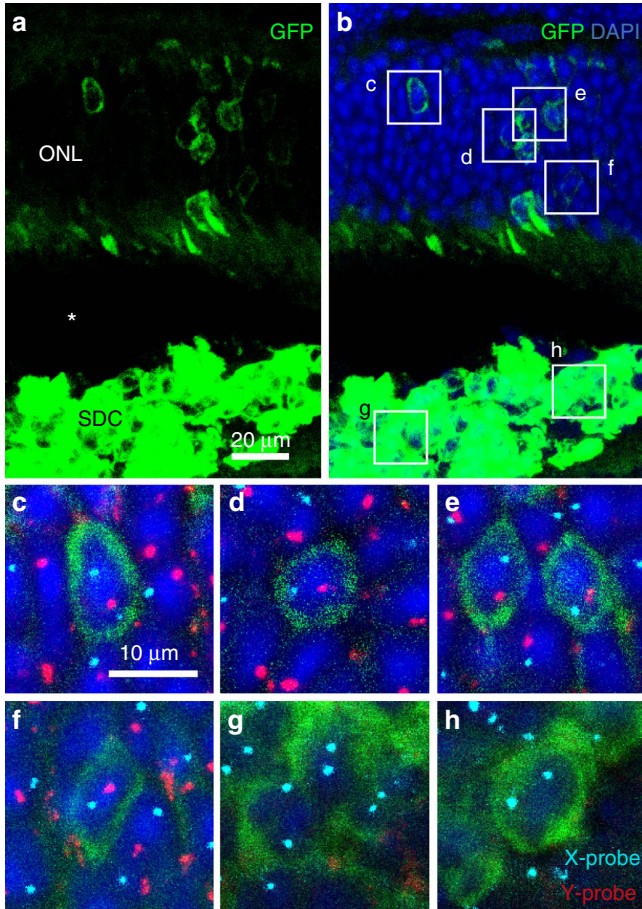

**Figure 2 | The nuclei of GFP-positive donor cells in the host ONL contain the Y chromosome that is only present in the host.** (**a,b**) Projection images showing donor photoreceptor precursor cells from P3 female *Nrl.GFP* mice that were transplanted into the subretinal space of an adult male wild-type host. The asterisk is a separation of the subretinal donor cells (SDCs) from the ONL due to a histological artefact. GFP-positive cells in the male host photoreceptor layer can clearly be seen to contain Y-positive nuclei (**c–f**). (**g,h**) GFP-positive cells in the subretinal donor cell mass contain only X chromosomes.

appearance of donor photoreceptor precursor cells that have 'integrated' into the host retina and have adopted the morphology of mature rod photoreceptors. Our data indicate that this may be a false conclusion in the majority of cases.

## Discussion

The observed mechanism of cell fusion in the absence of polyploidy may be explained by the merger of two separate lipid bilayer plasma membranes[14,15], resulting in the transfer of cytoplasmic contents between the fused cells. Our data support both homotypic (fusion between donor and host rods) and heterotypic cell fusion (between host rods and other donor retinal cells). We cannot currently rule out other mechanisms that might lead to fusion such as via membrane nanotubes[16], endocytosis or gap junctions[17]. Cell fusion, as a phenomenon that is well described and commonly reported in development, homeostasis, disease and regeneration[18], is a likely explanation of the mechanism observed here. Cell fusion is known to influence the cell cycle and so this process may increase the risk of neoplasm following photoreceptor transplantation[19,20]. With further understanding of cell fusion, this process may be harnessed as a therapeutic reprogramming mechanism[18,19] in future stem cell-based approaches in retinal regeneration and repair[21].

## Methods

**Mouse strains.** *Tg(Nrl-EGFP)* mice (herein *Nrl-GFP*) were a kind gift of A. Swaroop, Bethesda, MD, USA; mice that were homozygous for the *rd1* mutation (*Tg(CAG-DsRed*MST)1Nagy, Pde6b^rd1/rd1^*, Jackson Laboratories) were used for experiments and also crossed to C57BL/6J (Jackson Laboratories) wild-type mice to generate heterozygous DsRed animals that had a non-degenerate retina. *Gt(ROSA)26Sor^tm9(CAG-tdTomato)Hze^* mice (Ai9, Jackson Laboratories) were obtained locally (Ed Mann, University of Oxford, UK) and crossed to *Tg(Nrl-EGFP)* to obtain compound heterozygous pups for the Cre-loxP experiment. The *Tg(Crx-Cre)1Tfur* (ref. 22) mice were obtained from V. K. Yadav (Wellcome Trust Sanger Institute, Cambridge, UK) with the approval of the originator (T.Furukawa, Osaka University, Japan). This line provided hosts for the Cre-loxP experiments and was maintained by crossing *Tg(Crx-Cre)1Tfur* heterozygotes to C5BL/6J wild-type animals. The presence of the *Crx-Cre* transgene was identified by genotyping of ear biopsy tissue. The animals were maintained in the animal facility at the University of Oxford. All animal experiments were conducted according to the UK Home Office guidelines on the Animal (Scientific Procedures) Act of 1986 and were approved by the University of Oxford Animal Ethics Committee.

**Retinal dissociation and FACS.** The eyes were enucleated from postnatal day 4–7 mice and transferred to ice-cold 1× HBSS. The eyes were dissected and retinal

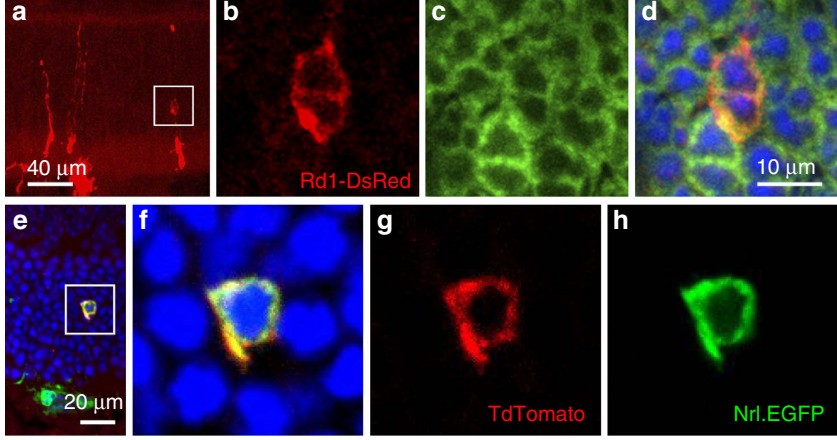

**Figure 3 | Further evidence of cytoplasmic transfer between donor and host.** (**a–d**) Donor cells collected from P1–3 DsRed mice, which were also homozygous for the *rd1* mutation were transplanted into adult *Nrl-GFP* mice in which rods are labelled with GFP. Three weeks after transplantation, a DsRed and GFP-positive photoreceptor cell bodies were seen. (**e–h**) To confirm cytoplasmic transfer of proteins using a Cre-lox recombination, donor cells (*CAG-LSL-tdTomato, Nrl-GFP*) at P7 were transplanted into adult *Crx-Cre* host mice. Three days post transplantation, donor cells in the subretinal space are GFP positive as expected, whereas in the host ONL (white box) a double-labelled (GFP and tdTomato) cell body is visible, confirming that Cre had passed from host photoreceptor to donor cell to activate tdTomato, which then retrogradely labelled the host photoreceptor together with GFP.

dissociation was performed using the Papain Dissociation System (Worthington Biochemical Corporation, Lakewood, NJ, USA) according to the manufacturers' instruction. The cells were re-suspended in 1% FCS in phosphate-buffered saline (PBS) at a concentration of $20 \times 10^6$ cells ml$^{-1}$. Propidium iodide (1 µg ml$^{-1}$, Sigma-Aldrich) was added before FACS sorting to exclude dead cells. GFP-positive photoreceptors were sorted using the Beckman Coulter Legacy MoFlo MLS High Speed Cell Sorter or Beckman Coulter MoFlo XPD cell sorter. Different emission filters and lasers were used for GFP (529/28 emission filter; 488 laser) and propidium iodide (625/26 filter; 561 laser). The sorted cells were collected in EBSS containing 10% fetal bovine serum and kept on ice. The cells were spun at 100 RCF for 20 min and the cell pellet was resuspended at a concentration of 200,000 cells µl$^{-1}$ in EBSS containing 0.005% DNase I and kept on ice before transplantation.

**Transplantation and tissue collection.** Host mice (5–8 weeks old) were anaesthetized using intraperitoneal ketamine hydrochloride (dose of 80 mg kg$^{-1}$ body weight) and Xylazine (dose of 10 mg kg$^{-1}$). Additional local anaesthesia was provided by proxymetacaine hydrochloride eye drops (0.5% m/v minims, Bausch & Lomb). The pupils were dilated using tropicamide (1% w/v minims, Bausch & Lomb) and phenylephrine hydrochloride (2.5% w/v minims, Bausch & Lomb). The cells were transplanted subretinally using a Hamilton syringe and a 34-gauge needle. The animals were recovered using Antisedan (Atipamezole, 2 mg kg$^{-1}$ bodyweight). At up to 3 weeks post transplantation, the animals were killed and the eyes were enucleated. Following removal of the lens, the eyes were fixed for 1 h in 4% paraformaldehyde in PBS. The tissue was processed as described by cryopreservation in sucrose before embedding in Tissue-Tek O.C.T Compound (Sakura, Alphen aan den Rijn, The Netherlands). Cryosections were cut (18 µm) using a Leica Cryostat and dried before storing at $-80\,°C$.

**FISH analysis.** Mouse bacterial artificial chromosomes (BACs) specific for the X (RP23-119M14 and RP23-168A19) or Y chromosome (BMQ 367K12, BMQ 451F0O8 and BMY53I13) were used as FISH probes, and labelled by nick translation (Abbot Molecular) according to the manufacturer's instructions, incorporating digoxigenin-11-dUTP (Roche) in the X probes and biotin-16-dUTP (Roche) for the Y probes. To suppress the hybridization of mouse repetitive DNA sequences, unlabelled mouse C0tI DNA (ThermoFisher Scientific) was added. Cryosections of retinas derived from sex-mismatched transplantations of GFP-positive donor cells in wild-type hosts were permeabilized in 0.5% Triton-X in PBS, blocked in 4% BSA in PBS-T (0.025% Tween-20) for 10 min at room temperature) and incubated in GFP booster Atto-594 (Chromotek, Germany) diluted 1:150 in 4% BSA PBS-T. Following washes in PBS-T, the sections were postfixed in 2% formaldehyde and incubated in 0.1 M HCl. The sections were rinsed in $2 \times$ SCC and 300–500 ng of FISH probes (25 ng µl$^{-1}$) were applied under a glass coverslip. The probes and tissue DNA were denatured at 85 °C for 8 min. The slides were incubated overnight at 37 °C in a humidified chamber. After hybridization, the slides were washed in $0.1 \times$ SSC at 60 °C. After a blocking step in 0.1% Tween-20, 4% BSA, the slides were incubated for 30 min at 37 °C in the secondary antibodies (anti-DIG FITC (Roche) and avidin-Cy5 (ThermoFisher Scientific). Following washes in 0.1% Tween20 in $4 \times$ SSC at 42 °C slides, the cells were counterstained in DAPI (250 ng ml$^{-1}$) in $4 \times$ SSC before mounting using ProLong Diamond Antifade mounting media (ThermoFisher Scientific). Microscopy (Zeiss LSM710 and Leica SP8 SMD X) was performed with sequential fluorescence excitation (excitation/collection wavelengths (nm): Hoechst 405/440–480; FITC (X) 488/494–550; GFP booster ATTO 594: 561/580–625; Avidin-Cy5 (Y) 633/650–700). Thin (0.8 µm) single confocal slices were used for co-localization studies. For all images, enhancement of brightness and contrast was performed using Zen Lite 2011 software (Zeiss). The images were pseudocoloured as follows: GFP booster Atto-594 in green, X probe in light blue, Y probe in red.

**Co-localization analysis.** Confocal microscopy was performed as described above. Thirty-four confocal slices (3–10 EGFP + host photoreceptors per image) were taken from three DsRed eyes that were transplanted with FACS-sorted *Nrl-EGFP* donor cells. The degree of co-localization of EGFP and DsRed was measured using the co-localization analysis programme in the ZEN software 2010. Mander's overlap coefficients (MOC) were computed for EGFP + cells within the host ONL and from a control experiment, that is, MOC = 0 (red cone arrestin staining on sections of adult *Tg(Nrl-L-EGFP)* mice. The P values were determined by unpaired Student's *t*-test.

**Data availability.** The authors declare that all data supporting the findings of this study are available within the article and all relevant data are available from the authors, on request.

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

## Acknowledgements

This work was funded by the Medical Research Council (UK), the Royal College of Surgeons of Edinburgh, the Special Trustees of Moorfields Eye Hospital, the UK Ministry of Defence, the NIHR Biomedical Research Centres at Moorfields Eye Hospital and Oxford University Hospitals NHS Foundation Trust, and the Wellcome Trust.

## Author contributions

R.E.M., M.S.S., A.R.B., J.B. and C.M.G. conceived the experiments. M.S.S., J.B., S.A.A., I.D., A.B.-C., A.R.B. and D.M. performed the experiments. R.E.M., M.S.S., J.B., D.M. and A.R.B. wrote the manuscript. All the authors contributed to data analysis and approved the final manuscript.

## Additional information

**Competing financial interests:** The authors declare no competing financial interests.

**Publisher's note**: 

