## [Peer Review File · Nature Communications]

Reviewer #1 (Remarks to the Author)

In their paper "Cell Fusion in Photoreceptor Transplantation" Singh et al. describes an important finding in cell transplantation to mouse retina. Namely that most photoreceptors do not migrate and integrate into the retina but instead exchange cytoplasmic proteins with the existing photoreceptors. This finding is very important for clinical studies of cell transplantation. The experiments are performed at a high standard and the results are clear.

I have only one recommendation: please include more quantification of the results. Otherwise I recommend accepting the paper for publication.

Reviewer #2 (Remarks to the Author)

In their manuscript titled "Cell fusion in Photoreceptor Transplantation," Singh et al. explore the possibility that the cells that adopt the morphology of normal photoreceptors after transplantation of GFP labeled rods may be the endogenous photoreceptors that have received some component of the transplanted cells (DNA, RNA, protein etc...). First, they transplant Nrl-GFP cells into a constitutive dsRED recipient and find abundant co-localization of GFP and RFP. To confirm this finding, they also transplanted female GFP+ photoreceptor precursors into adult wild type male recipients. Using FISH, they identified abundant GFP positive cells with a Y chromosome. These data suggest that the nuclei do not fuse in the transplantation experiments. They also transplant total retinal cell lysate from Rd1 dsRED mice at P1-P3 into Nrl-GFP recipients and found double positive photoreceptors 3 weeks later. They interpreted this experiment to suggest that cells other than photoreceptors may be the source of transfer of the RFP gene/transcript or protein. A Cre reporter strategy was used to demonstrate bi-directional transfer of information between donor and recipient photoreceptors.

I do not agree with the interpretation of the Rd1 transplantation experiment. There are clearly photoreceptor precursors in the transplanted cells and they could have transferred their dsRED encoding information shortly after transfer and before the rods died. The text needs to be modified and more clearly state this possibility and the limitations of the experiment performed.

The problem with the Cre experiment is that they only looked 3 days after transplant. They concluded bi-directional exchange of DNA/RNA/Protein but in fact that was not demonstrated. The double positive cell may simply be a donor cell that received Cre from the recipient photoreceptors. They would need to look at 3 weeks and identify a cell in the ONL with mature photoreceptor morphology to support the claim in the paper.

Point by point response – Singh et al. Cell fusion in photoreceptor transplantation

1. Reviewer #1 – The quantification experiments were designed using the Mander's overlap coefficient, which is detailed on lines 60-62 and in figure 1. The tissue processing and cell counts in these experiments were designed specifically for quantitative analyses. We have now also added further quantification of the number of double labelled cells after Cre-lox recombination on line 103.

2. Reviewer #2 – We agree with the reviewer that our interpretation that no rods could be transplanted in the rd1 mouse donor experiment could be challenged, since the fusion itself may alter the properties of the transplanted cell differentiating in the subretinal space. One possibility is that the missing enzyme (beta-6 PDE) could transfer from the host cells to the transplanted rd1 cells and thereby sustain them for longer, allowing these cells to differentiate into rods. We thank the reviewer for this insight and we have amended our comments on this in lines 86-88:

“Alternatively, it is also possible that donor rod precursors could have been rescued from degeneration by the cytoplasmic transfer of beta-6 PDE in the reverse direction of the DsRed - from the host rod outer segments into the maturing rd1 mouse donor cells.”

3. Reviewer #2 – The reviewer comments that “the double positive cell may simply be a donor cell that received Cre from the recipient photoreceptors.” We agree with this and indeed it was the basis for performing the experiment – it confirms protein transfer (Cre) in the opposite direction, from host back to donor cell. Since the green fluorescent protein is in the donor cell and needs to travel in the opposite direction, the presence of both labels implies bidirectional transfer. Figure 3e shows a fully integrated double labelled photoreceptor deep within the host outer nuclear layer – in this case both markers originate from the subretinal cell, but the host cell would have first needed to transfer Cre to it, in order for the red tdTomato transgene to be activated.

4. Reviewer #2 – With regard to looking later for cre-lox recombination, we deliberately looked as early as possible in these experiments, because we wished to identify the earliest point at which fusion occurred with bidirectional transfer between two cells. If we had looked at later time points as well, we could potentially pick up cells that had been activated by an earlier fusion event with one cell, but then initiated a second fusion event with another. This is because the cre-lox event only needs to occur once for the transplanted cell to express tdTomato indefinitely. This red fluorescent marker would therefore be transferred to any host cell that it might subsequently fuse with. Making conclusions about cre-lox recombination at later time points therefore assumes that the fusion event is singular and long-lasting, which might not be the case. We have therefore toned down our conclusions on this result on lines 101 to 108, in light of the reviewer's suggestion:

“However the number of the GFP+ host cells that also contained tdTomato was small, suggesting that the fusion event may not be stable. This is because an undetermined time would be required for the Cre-lox recombination event, whereas the GFP transfer could occur almost spontaneously after cytoplasmic fusion. It cannot however be excluded that this occurred by two successive fusion events, with cre being taken up from one photoreceptor and subsequent fusion and retrograde labelling of another.”